# Polymeric Nanocomposite Hydrogel Scaffolds in Craniofacial Bone Regeneration: A Comprehensive Review

**DOI:** 10.3390/biom13020205

**Published:** 2023-01-19

**Authors:** Maha H. Bashir, Nahed S. Korany, Dina B. E. Farag, Marwa M. S. Abbass, Bassant A. Ezzat, Radwa H. Hegazy, Christof E. Dörfer, Karim M. Fawzy El-Sayed

**Affiliations:** 1Oral Biology Department, Faculty of Dentistry, Cairo University, Cairo 11553, Egypt; 2Stem Cells and Tissue Engineering Research Group, Faculty of Dentistry, Cairo University, Cairo 11553, Egypt; 3Clinic for Conservative Dentistry and Periodontology, School of Dental Medicine, Christian Albrechts University, 24105 Kiel, Germany; 4Oral Medicine and Periodontology Department, Faculty of Dentistry, Cairo University, Cairo 11553, Egypt

**Keywords:** nanohydrogels, craniofacial defects, maxillofacial defects, tissue engineering

## Abstract

Nanocomposite biomaterials combine a biopolymeric matrix structure with nanoscale fillers. These bioactive and easily resorbable nanocomposites have been broadly divided into three groups, namely natural, synthetic or composite, based on the polymeric origin. Preparing such nanocomposite structures in the form of hydrogels can create a three-dimensional natural hydrophilic atmosphere pivotal for cell survival and new tissue formation. Thus, hydrogel-based cell distribution and drug administration have evolved as possible options for bone tissue engineering and regeneration. In this context, nanogels or nanohydrogels, created by cross-linking three-dimensional polymer networks, either physically or chemically, with high biocompatibility and mechanical properties were introduced as promising drug delivery systems. The present review highlights the potential of hydrogels and nanopolymers in the field of craniofacial tissue engineering and bone regeneration.

## 1. Introduction

Craniofacial bones bear the masticatory forces, protect the brain and eyeballs as well as support the vital structures of the head [1]. Bone possesses a complex composition with a dense outer cortical bone, built up of repeating osteon units, and a porous cancellous spongiosa core, made up of an interconnected framework of trabeculae with bone marrow-filled spaces. Collagen (COL) fibers and calcium phosphate crystals make up the trabeculae and osteon units [2]. Bone development takes place within a structured COL matrix, with fiber bundle diameters ranging from 50 to 500 nm, a 67 nm periodicity and 40 nm gaps between COL fibers [3,4,5,6]. Hydroxyapatite (HAp) crystals become embedded in the spaces between COL fibers, increasing the bone’s stiffness [7,8]. The structure and organization of HAp crystals, extracellular matrix (ECM) components and cells together determine the qualities of bone tissues [9]. Defects caused by trauma, tumor or cyst, resection, infectious diseases and congenital or developmental conditions may result in serious functional, aesthetic and psychological sequelae [10]. In such conditions, the absence of hard and soft tissues leads to disfigurements that compromise basic functions, including mastication, speech and swallowing, in addition to limited thermal and physical protection of important anatomical structures [11,12,13].

Especially, the existence of critical size bony defects [14], not expected to heal spontaneously, poses a challenging clinical situation. In grafting such defects, autogenous tissues have long been regarded as the gold standard [15]. Autografts possess all essential ingredients required to achieve successful tissue regeneration, including cells, growth/differentiation factors and ECM components [16]. Yet, aside from the commonly limited amounts of tissues available for harvesting to adequately fill or cover a significant maxillofacial defect, using autogenous tissue necessitates collecting it from a donor, with constrains regarding the surgical procedure, time, patient’s morbidity and possible complications [17]. Thus, alternative materials, including allografts, xenografts and alloplasts, have been introduced in clinical practice to circumvent these constraints [18], serving primarily as osteo-inductive and/or conductive scaffolds, supporting the migration of cells from the periphery into the grafted area [19,20,21].

The field of bone tissue engineering (TE) is structured around four key components; firstly, osteogenic cells; secondly, a scaffold created using bioactive materials that mimic the bone ECM; thirdly, vascularity that provides adequate transport of nutrients; and finally, morphogenetic signals to direct the new tissue formation [22,23] (Figure 1). In this context, nanocomposite biomaterials represent a new class of materials that combine a biopolymeric and biodegradable matrix structure with bioactive and easily resorbable nanosized fillers [24]. The nanofillers introduced into such a polymeric matrix confer important physical and chemical properties to the biomaterial, including increased surface area, improved mechanical strength and stability, improved cell adhesion, proliferation and differentiation [24,25]. This review article provides an overview on the usage of nanoparticles (NPs) incorporated into polymers or nanopolymers in the form of hydrogels in the field of bone TE, exploring their biomimetic properties, mechanical strength and biocompatibility.

## 2. Hydrogel Scaffolds in Bone Regeneration

Hydrogel-based cell and drug administration have been proposed as promising agents in the field of bone TE and regeneration. In addition to their physical plasticity with the ability to adapt to any required shape during injection or implantation, they are postulated to create a three-dimensional (3D) natural hydrophilic atmosphere that promotes cell survival and development. Moreover, their degradation rate, porosity and release profile can be readily regulated by modifying the technique and degree of cross-linking [26].

Ideally, a hydrogel in the field of bone tissue regeneration should be non-cytotoxic, non-immunogenic, osteoinductive, osteoconductive, osteogenic, imitate the ECM in nature with suitable pore size, enhance cell adhesion, proliferation and osseointegration and allow nutrients, oxygen and metabolic waste to circulate at the implant site. It should further be degradable by hydrolysis or endogenous enzymes in synchronization with new bone ingrowth, creating sufficient room for new bone development, allowing the release of encapsulated bioactive substances, be structurally stable with adequate mechanical strength to be utilized in load-bearing areas and, lastly, possess the capacity to be injected to ease the process of administration [27]. It is generally believed that loading stem cells and growth factors in the hydrogel matrix should boost the rate of new ECM synthesis [28]. Yet, a fast hydrogel breakdown before the creation of a new ECM [29] should be avoided, in addition to ensuring the appropriate direction of stem cell differentiation, which remains essential to guarantee appropriate repair and regeneration [30]. Therefore, developing controllable hydrogels with good mechanical stability and prolonged release is critical in developing an effective therapy for bone repair/regeneration [26].

It is noteworthy that various natural and synthetic polymeric-based hydrogels have been designed for drug delivery. The combination of precise chemistry with multifunctional materials leads to unique responsive versatile hydrogels, which can be employed as a potential platform to facilitate advanced biomedical applications. For example, amphiphilic linear pentablock hybrid polypeptides of the ABCBA type were synthesized using precise chemistry, where A is poly(L-lysine), B is poly(L-histidine)-co-poly(γ-benzyl-L-glutamate) and C is poly(ethylene oxide). The blocks’ chain lengths were changed in order to produce hydrogels with various viscoelastic characteristics. An extrudable, in situ-forming, very quickly self-healing hydrogel with responsiveness to pH, temperature and enzymes was produced. These characteristics would render the hydrogel appropriate for the directional and targeted delivery of cargo from the hydrogel toward cancer tissues in drug delivery since these tissues have lower pH and higher temperatures [31]. On the other hand, delivery systems that can safeguard various drugs, including osteoinductive growth factors from degradation, manage their delayed release to the intended site and moderate their biological action for an extended time, are required for the treatment and regeneration of damaged bone tissue [32].

### 2.1. Classification of Hydrogels Used in Bone Regeneration

Hydrogels can be generally categorized according to their origin, production technique, cross-linking characteristics, distribution mechanism and degradability [33].

#### 2.1.1. According to Origin

##### Natural Hydrogels

Natural proteins (e.g., fibrin, fibroin, COL), and gelatin (Gel) and polysaccharides (e.g., chitosan (CS), hyaluronan and alginate (AG)) can be used to synthesize hydrogels. Being components of the natural ECM, natural polymers are highly bio-compatible with a minimal immunological response. They enhance cell adhesion and proliferation, promote tissue regeneration as well as provide mechanical stability and structural integrity to bone structures. Yet, their principal disadvantages lie in their weak mechanical capabilities as well as their rapid degradation under certain biological environments, depending on the level of certain enzymes and the location of their implantation [34,35].

COL is a bioactive natural polymer that exhibits favorable biocompatibility, facilitating adhesion and proliferation of bone cells, with decreased antigenicity [36,37]. The bone’s ECM is primarily composed of COL [38]. Yet, the faster biodegradability rate of pure COL scaffolds, poor mechanical strength and increased swelling potential persuaded instead the use of COL-based composite biomaterials for bone TE [39,40,41,42]. AG, further a naturally occurring anionic polymer made up of 1,4-linked-D-mannuronic acid and -L-guluronic acid, is typically obtained from brown seaweed and has been extensively investigated and used in many biomedical applications, relying on its biocompatibility, low toxicity, relatively low cost and simple gelation potential through the addition of divalent cations such as calcium (Ca^2+^) [43,44,45].

##### Synthetic Hydrogels

Biodegradable polymer compounds, such as polyethylene glycol (PEG), polyvinyl alcohol (PVA), poly (lactic acid), polyacrylamide (PAM) and their copolymers, can be utilized to synthesize hydrogels for bone repair and regeneration, transporting active proteins, growth factors and medicines [46]. Synthetic polymers are characterized by a high mechanical strength, a low cost and customized properties appropriate for specific applications [47,48]. Their chemical composition and ratio of the above-mentioned polymers determine their characteristics, including porosity, degradation time and mechanical features [49]. These polymers could be manufactured in large amounts in sterile environments, ensuring homogeneous and repeatable characteristics without the risk of immunogenicity or infection. However, the acidic nature of their breakdown products results in unfavorable local pH shifts. Moreover, their hydrophobic nature makes it difficult for cells to adhere to them, which worsens their osteoinductive properties in the field of bone TE [50].

##### Natural and Synthetic Polymer Hydrogels (Composite Hydrogels)

A combination of natural polymers with synthetic ones was proposed to obtain improved physicochemical characteristics of both classes [51]. Cross-linking hydrogel films of pectin-grafted acrylamide with glutaraldehyde demonstrated superior film formation, gelation and mechanical qualities compared to pure pectin [52]. Chemical, photo and gamma-ray initiations were used to produce cellulose-supported synthetic polymerizable monomer hydrogels [53]. Further hydrogel formulations have been described, including AG, alginic acid (ALG), carrageenan, arabic gum and xanthan gum modified with synthetic polymers and synthetic polymerizable monomers by various processes [54].

#### 2.1.2. According to Their Production Technique

##### Microbead Hydrogels

Microbeads (MBs) have shown excellent potential in the encapsulation of living cells and drugs, relying on their cross-linking techniques. An injectable MB AG hydrogel was produced to encapsulate mesenchymal stem cells (MSCs) derived from dental origin, such as periodontal ligament stem cells (PDLSCs) and gingival mesenchymal stem cells. For ionic cross-linking, the AG and stem cell mix were injected with a syringe into a calcium chloride solution. Due to effective nutrition and oxygen transfer, ectopic mineralization was evident both inside and outside the MBs, while maintaining the cellular vitality [55]. Additionally, MBs of polymer hydrogels, produced by microfluidics techniques, emulsifying agents, electrostatic droplets extrusion, coaxial air jetting and in situ polymerization, were utilized in the field of bone regeneration and repair. To produce uniform MBs, recently, a non-equilibrium microfluidic technology was introduced for the production of smaller-sized hydrogel beads (<100 mm) [56]. Still, further research is needed to develop biocompatible, osteoconductive, osteoinductive and osteogenic hydrogel MB formulas [57].

##### Fibrous Hydrogels

Nanofiber was reported to sustain the release of incorporated growth factors, where the release profiles of loaded proteins could be adjusted by altering nanofiber concentrations [58,59]. Fibrous hydrogels have diameters ranging from a few nanometers to a few microns [60]. Two stages are typically required to create hydrogel fibers, including spinning and cross-linking procedures. Spinning techniques involve electrospinning [61], microfluidic spinning [62], wet spinning [63], gel spinning [64], hydrodynamic spinning [65] and 3D printing technology [66]. To finally synthesize the hydrogel fibers with the required properties, additional cross-linking with glutaraldehyde, enzymes, or thermal or ultraviolet radiation is required [62]. Because of their high surface-to-volume ratio, quick reaction and immobilization capabilities, hydrogel fibers have shown great potential in the field of TE [67]. Unfortunately, these fibers have certain drawbacks, including poor mechanical qualities and rapid release, that must be overcome to create a hydrogel formula that allows for controlled, sustained release of proteins and drugs delivery [57].

#### 2.1.3. According to Cross-Linking

Hydrogels can be further divided into chemical hydrogels, physical hydrogels and a mix of both (hybrid) according to the kind of cross-linking forces between the polymeric chains in the hydrogels [68].

##### Physically Cross-Linked Hydrogels

Physical gels are highly water soluble and thermo-reversible. In a physiological medium, this form of hydrogel has a short lifetime, ranging from a few days to a month at most. As a result, physical gels are employed, where rapid medication release is necessary. Physical hydrogels can be formed by ionic complexation of CS with tiny anionic molecules such as sulfates, phosphates, citrates or anions of platinum, palladium and molybdenum [69].

##### Chemically Cross-Linked Hydrogels

Chemically cross-linked hydrogel networks are easier to manage than physical hydrogel networks since their manufacture and usage are not pH-dependent. Chemically cross-linked hydrogels are called permanent hydrogels, as they do not dissolve in the surrounding media due to the presence of strong covalent bonds between the macromolecular chains [68]. Superior thermal, mechanical, chemical and surface characteristics are provided by chemical cross-linking to the prepared hydrogels [70].

##### Hybrid Hydrogels

Hybrid hydrogels are complex structures composed of building units different from each other morphologically, functionally and chemically, connected together by physical or chemical means. They can be formed from hundreds of physically or chemically cross-linked nanohydrogels (NGs), or they are composed of different NPs, pro-polysaccharides and/or polysaccharides [71] and/or polymers, such as organic carbon, magnets and plasmon NPs [72]. Based on the size and function of these fundamental components, hybridization can occur at the microscopic or molecular level [73,74].

NPs can generally be divided into four groups: metallic, carbon, polymeric and plasmon. Each category has a unique set of characteristics that makes it suitable for a certain biomedical specialty [75]. For example, metallic-based hydrogel nanocomposites can be remotely controlled and are responsive to electric/magnetic field stimuli. They are frequently utilized as imaging agents, conductive scaffolds, actuators/sensors and drug delivery systems. They also exhibit antimicrobial capabilities [76]. The major usage of polymeric-based hydrogel nanocomposites, which may be pH, temperature, concentration or light sensitive, is in the field of controlled drug delivery [77].

Carbonaceous nanomaterials (CNMs) have great potential among the nanomaterials created with the development of nanotechnology and nanomedicine. CNM-based nanocomposites exceed other nanomaterials (metal, organic, etc.) in terms of surface immobilization of macromolecules (such as proteins, enzymes, peptides, etc.), biocompatibility, mechanism of sensing, rapid transfer of electron kinetics ability, heat transfer and surface adsorption ability thanks to their distinctive architectures, substantial surface area, ability to overcome biological barriers and impressive physicochemical characteristics. With the use of laser ablation, carbon vapor deposition, arc discharge and joule heating, CNMs, which include fullerene, carbon nano-onions, carbon dots, graphene, graphene oxide and reduced graphene oxide, are primarily manufactured [78].

Due to the strength qualities of carbon nanotube surfaces (CNTs), noncovalent interactions between polymer and CNT surfaces result in hybrid nanocomposite materials. Single wall carbon nanotubes (SWCNTs) in particular have been employed in biomedical fields such as gene therapy, medical image processing, medication delivery, TE and other fields [79]. A uni-compartmental knee implants with ultra-high molecular weight polyethylene sheets reinforced with functionalized SWCNTs with concentrations of 0.01 and 0.1 wt% was fabricated. The produced hybrid nanocomposite material samples exhibited improved yield strength, tensile strength, elongation, Young’s modulus and, after 14 days of incubation, human osteoblast cells demonstrated improved cell viability along with great cell growth and differentiation. Such combinations confirm the value of using CNTs for biomedical applications, providing an excellent potential for the creation of innovative composite biocompatible materials [79].

The mechanical strength of inorganic-based hydrogel nanocomposites makes them popular for usage in bone-related implantations [77]. In the same context, HAp is the most commonly used NP due to its close resemblance to the inorganic component of the natural bone matrix [80]. It is considered to be the optimal candidate for bone repair due to its excellent biocompatibility, osteoconductivity and non-toxic nature [81]. Moreover, HAp NPs have been utilized as carriers for growth factors, owing to their excellent capacity for adsorbing proteins [82]. As HAp has low mechanical properties with low biodegradability and no osteoinductivity, intensive efforts have been made to introduce a variety of ionic substitutions such as silicon (Si^4+^), magnesium (Mg^2+^), zinc (Zn^2+^), fluoride and carbonate ions, into the apatite structure of HAp at different positions [83,84,85]. In this context, the introduction of the bivalent cation of strontium (Sr^2+^) showed an interesting potential to stimulate bone formation and inhibit bone resorption [86].

#### 2.1.4. Smart Hydrogels

To overcome the shortcomings of currently-present natural or synthetic hydrogels, certain chemical changes were made to build smart hydrogel systems, with enhanced mechanical properties and improved biocompatibility, ideal for TE of bone and cartilage [57]. These include the interpenetrating polymer networks, double networks, shape memory and self-healing hydrogels, programmable hydrogels and 3D printed hydrogels.

##### Interpenetrating Polymer Network Hydrogels

An interpenetrating polymer network (IPN) is a polymer composed of two or more networks that are interconnected in some part at the molecular level but are not covalently linked to each other and cannot be detached until chemical connections are broken. They are considered to be a variety of hybrid polymers with exceptional physio-chemical properties [68]. Since the IPNs are cross-linked, they swell but do not dissolve in the presence of solvents; moreover, creep and flow can be nearly suppressed. Therefore, the mechanical strength and elasticity of the material is enhanced, besides the physical and chemical characteristics, including temperature sensitivity and interfacial compatibility [87]. IPNs and semi-IPNs (if only one IPN polymer is cross-linked, while the other is in a linear form, or a polymer contains both linear and branched polymers) have evolved as novel TE hydrogels, with enhanced mechanical characteristics, permitting cell attachment and proliferation. Combining them to obtain the advantageous qualities of each polymeric component (IPNs or semi-IPNs) could generate a new material with completely different properties [57].

##### Double Network Hydrogels

Double network (DN) hydrogels possess extraordinary mechanical strength and toughness due to their unique contrasting network structures, strong interpenetrating network entanglement and efficient energy dissipation [88]. Because they can be cross-linked in situ, injectable DN hydrogels were created to preserve cells’ integrity during injection. Carbohydrate polymers are one of the most commonly employed materials used in the production of such injectable in situ hydrogels, being readily accessible and biocompatible, with adjustable functional component [89].

##### Shape Memory and Self-Healing Hydrogels

Shape memory (SM) and self-healing (SH) hydrogels provide a unique method to avoid the limitations of injectable hydrogels, including gelation time, biomechanical compatibility and tissue functionality, while preserving the original defect morphology and retaining cell viability [57]. Tensile strength of SM hydrogels was reported to be 2.3 MPa, while SH hydrogels demonstrated a tensile strength of 0.7–1.7 MPa. However, because thermally induced SM hydrogels rely on heat to activate their self-healing abilities, their field of usage is very limited in the biomedical sector [35].

##### Programmable Hydrogels

Programmable hydrogels can change their characteristics and functionalities selectively and/or successively. They can experience functional changes in size, mechanical support, cell adhesion and molecular sequestration when stimulated. They represent stimuli-responding hydrogels whose transformations are passive and cannot be arrested or reversed once initiated. Their current functions are satisfactory if they are utilized to isolate target cells from a cell mix in vitro or to aid in the understanding of specific fundamental biological processes. Yet, numerous technological issues must be taken into consideration if they are produced for in vivo cell homing and protein delivery [90].

##### Three-Dimensional Printed Hydrogels

3D bioprinting is a new TE technique in which a computer-aided biocompatible material is constructed layer upon layer using biological ink while incorporating cultivated cells in the 3D printed scaffolding. Since they have many characteristics identical to the natural ECM and may also offer a highly hydrated environment for cell growth, these hydrogels represent promising biomaterials [91]. The micro-extrusion method is the most commonly used to create constructions consisting of 3D hydrogels enclosing cells, including chondrocytes or stem cells [92,93]. In this context, 3D-printed cryogel or hydrogel moldings with porosities extending from 79.7% to 87.2% and strong interconnectivity were utilized to produce patient-specific TE scaffolds in craniofacial cleft defects treatment [94]. To enhance medium percolation during 3D printing and, in turn, the proliferation of the cells attached to the material, the porosity of the material becomes crucial. Because of their superior biodegradability and biocompatibility, polyhydroxybutyrate (PHB)-based nanocomposites are frequently employed in both TE and drug delivery systems. However, due to insufficient physicochemical and mechanical qualities, PHB’s utility in bone tissue engineering is restricted. Recently, PHB-based nanocomposites using a nanoblend and nano-clay with organically modified montmorillonite (MMT) were fabricated. The blended (PHB/MMT) has shown promise in 3D organ printing, lab-on-a-chip scaffold construction and bone TE [95].

## 3. Nanohydrogels

NGs are composed of diverse types of polymers of synthetic or natural origin. Their combination is bound by chemical covalent bonds or is physically cross-linked via non-covalent bonds, including electrostatic interactions, hydrophobic interactions and hydrogen bonding. Their remarkable ability to absorb water or other fluids is mainly attributed to their expressed hydrophilic groups, as hydroxyl, amide and sulfate [96,97]. NGs exhibit several pivotal hydrogel qualities, including high biocompatibility and mechanical capabilities, and are extremely useful in bone regeneration applications. NGs possess the characteristics of NPs with sizes ranging from 1 to 100 nm [97,98].

In contrast to conventional hydrogels (macro or microgels), which generally involve intermolecular cross-linking, intramolecular cross-linking is predominant in NGs [99]. Due to their smaller size compared to macro or micro hydrogels, NGs can better retain incorporated drugs [100]. Thus, NGs are being investigated for their delivery of bioactive substances at a rapid pace due to their several advantages over conventional hydrogels in terms of longer half-life, better loading capacity, superior encapsulation stability and superior tissue uptake [101,102,103,104]. NGs could transport proteins without causing denaturation, using in situ encapsulation approaches, during which the proteins are encapsulated in situ in the process of NG formation [98]. Moreover, NGs possess better suitability for the parenteral route of administration, as they can move inside the fine capillaries due to their small size. All these factors make NGs a preferred choice for the delivery of bioactive compounds over conventional hydrogel systems [99,105,106,107,108] (Figure 2).

### 3.1. Nanohydrogels in Cranio/Maxillofacial Regeneration

Similar to the Conventional Hydrogels, NGs can be categorized based on their source and origin into natural, synthetic and composite polymeric network-based NGs (Figure 3). Natural and synthetic polymer-based NGs can be created using a variety of methods, including chemical, ionic, self-assembly, electrostatic, reverse mini-emulsion, hydrophobic and micelle cross-linking. Novel and distinctive NGs biomaterials were produced through altering the chemical composition, the synthesis approach and cross-linking designs (cross-linking method, cross-linking agents) [109]. In this section, the NGs employed in the field of cranio- and maxillofacial tissue regeneration will be discussed in detail.

#### 3.1.1. Natural Polymer-Based Nanohydrogels (Table 1) (Figure 4)

Table 1 and Figure 4 are shown below.

**Figure 4 biomolecules-13-00205-f004:**
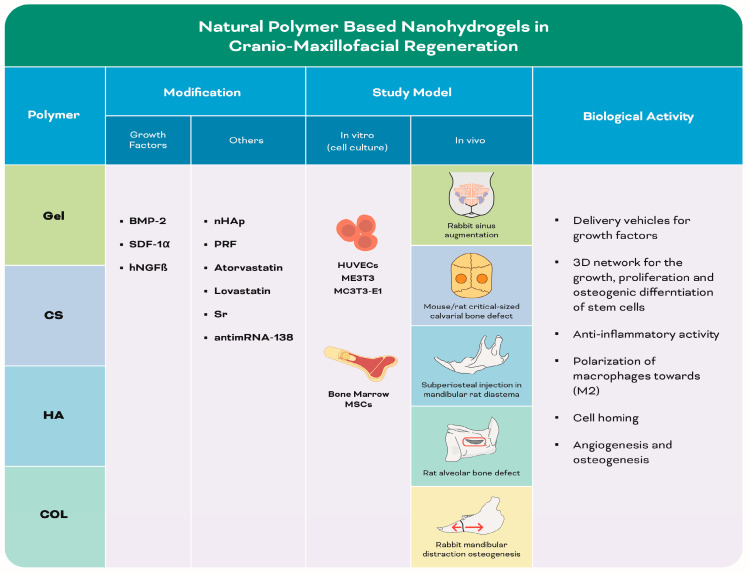
Infographic illustrating experimental studies of natural polymer-based nanohydrogels in Table 1. The red box refers to the alveolar bone defect. The red arrows refer to the mandibular distraction.

**Table 1 biomolecules-13-00205-t001:** Summary of the included studies employing different natural polymer-based nanohydrogels in cranio-maxillofacial regeneration.

Author, Year	Polymer	Co-Polymer	Modification	Main Features of Nano-Polymer	Study Model		Biological Activity	Outcomes
					In Vitro (Cell Culture)	In Vivo		
Mu et al., 2020 [110]	Gel		iPRF	Double network	HUVECs	Rabbit sinus augmentation	GelNPs acted as delivery vehicles for sustained release of growth factors from iPRF	GelNPs-iPRF composite enhanced bone regeneration after 8 weeks
Patel et al., 2020 [111]	CS	GG, ALG, KCA	Incubation in simulated body fluid promoted mineral deposition (mineralized hydrogel)	Fibrous hydrogel. Single fiber exhibited periodic regions at nanoscale		Mouse critical-sized calvarial bone defect	Sulfate group in CS-KSA improved bone regeneration by binding to proteins	CS-KCA mineral and non-mineral hydrogel significantly enhanced bone regeneration after 12 weeks
Mi et al., 2017 [112]	CS/GP	CMCS- NPs	SDF-1α	Crosslinked network with regular pores		Rat critical-sized calvarial bone defect	SDF-1α induced osteogenic differentiation of MSCs	SDF-1α/CS/CMCS- NPs embedded CS/GP hydrogel significantly increased new bone formation after 8 week
Wu et al., 2018 [113]	CS/GP		CS-T-HA/antimiRNA-138 NPs, SDF-1α	Porous structure	Bone marrow MSCs	Rat critical-sized cranial defect	Dual release of SDF-1α and CS/antimiRNA-138 from NPs promoted cell homing and osteogenic differentiation of MSCs	SDF-1α/NPs/hydrogel enhanced bone regeneration after 8 weeks
Petit et al., 2020 [114]	CS		Atorvastatin nanoemulsion, Lovastatin nanoemulsion			Mice calvarial bone defect	Thermosensitive hydrogel controlled the release of atorvastatin and lovastatin, inducing anti-inflammatory and osteogenic activity	Chitosan gel loaded with atorvastatin or lovastatin significantly improved bone regeneration after 2 weeks
Ding et al., 2019 [115]	CS	Dextran	Sr-nHAp	3D porous structure	MC3T3-E1	Rat critical-sized calvarial bone defect	The Sr caused polarization of macrophages towards (M2) phenotype and facilitated osteogenic differentiation of stem cells	Sr100nHAp/CSD hydrogel enhanced bone regeneration after 8 weeks
Martínez-Sanz et al., 2012 [116]	HA		nHAp, BMP-2			Subperiosteal injection in mandibular rat diastema	-Osteogensis and angiogenesis were directly correlated with the amount of BMP-2. -nHAp and BMP-2 functioned synergisticly to enhance hydrogel osteogenic activity	HA-based hydrogels containing nHAp and BMP-2 achieved mandibular bone augmentation after 8 weeks
Pan et al., 2020 [117]	CS	HA	nHAp	Porous structure with the nanoparticlesdispersed uniformly in the hydrogel system	ME3T3	Rat alveolar bone defect (tooth extraction)	-The hydrogel provided a 3D surface for the growth, proliferation and differentiation of stem cells -Decomposition of loaded nHAp produced a high concentration of calcium and phosphorus that stimulated osteogenic differentiation of stem cells	Hydrogel-nHAp composite scaffold demonstrated accelerated alveolar ridge preservation after 4 weeks.
Cao et al., 2012 [118]	COL	AG	nHAP, hNGFβ			Rabbit mandibular distraction osteogenesis	hNGF was protected and was able to retain its biological activities	hNGFβ in COL/nHAp/AG hydrogel enhanced bone regeneration after 14 days

Gelatin: Gel; injectable platelet-rich fibrin: iPRF; human umbilical vein endothelial cells: HUVECs; nanoparticles: NPs; chitosan: CS; gellan gum: GG; alginic acid: ALG; kappa carrageenan: KCA; β-glycerol phosphate disodium salt: GP; carboxymethyl chitosan: CMCS; stromal cell-derived factor-1α: SDF-1α; mesenchymal stem cells: MSCs; tripolyphosphate: T; hyaluronic acid: HA; strontium: Sr; nanohydroxyapatite: nHAp; three-dimensional: 3D; murine pre-osteoblast cell line: MC3T3-E1; bone morphogenetic protein: BMP; mouse calvaria-derived (subclone 14) osteoblast-like cells: ME3T3; collagen: COL; alginate: AG; human nerve growth factor beta: hNGFβ.

##### Gelatin-Based Nanohydrogels

Gel is a mixture of peptides and proteins taken out of the skin, bones and connective tissues by partial hydrolysis of COL [119]. Owing to its biocompatibility, biodegradability, hydrophilicity and cost-effectiveness, Gel is widely used in pharmaceutical and biomedical applications and as a TE scaffold [120,121]. A bioactive hydrogel based on autologous injectable platelet-rich fibrin (iPRF) modified with Gel NPs was developed to improve the mechanical strength and delay biodegradation with sustained release of iPRF-entrapped growth factors [122,123]. This natural hydrogel demonstrated a DN structure that enhanced the mechanical properties of the hydrogel, including injectability, toughness values and degree of deformation. Following four and eight weeks, implantation in a sinus augmentation model, sinus cavities treated with Gel NPs-iPRF hydrogels demonstrated a significantly higher amount of new bone formation as compared to Gel NP gels and the empty control histologically and radiologically, with more pronounced angiogenesis. The newly-formed bone gradually matured and remodeled into lamellar bones after eight weeks [110]. The upregulated bioactivity and osteogenic properties demonstrated by Gel NP-iPRF hydrogels were attributed to the capacity of the Gel NPs within the DN gels to absorb protein-based growth factors by forming electrostatic interactions or hydrogen bonds between proteins and Gel macromolecules [124,125,126,127], thus acting as a delivery vehicle for sustained release of bioactive components from iPRF.

##### Chitin/Chitosan-Based Nanohydrogels

Chitin (CN) is a unique co-polymer extracted primarily from shellfish sources, composed mainly on the *N*-acetyl-glucosamine monomer and *N*-glucosamine units. CN, together with its alkaline deacetylated derivative CS, are useful candidates for TE due to their non-toxic, biodegradable and biocompatible nature [128]. CS is a hydrophilic, cationic linear polysaccharide. Hydrogels made from CS contain D-glucosamine and *N*-acetyl-D-glucosamine units, which are -(1,4) connected [129]. Additionally, CS has proven to be a safe and cost-effective delivery system for proteins including stromal cell-derived factor-1α (SDF-1α), fibroblast growth factor (FGF), bone morphogenetic protein-7 (BMP-7) as well as genetic materials and anti-cancer agents [130,131].

To overcome the limitations of COL-based hydrogels grafts, including failure to provide structural guidance to native cells due to their fast degradation rate, bioinspired hydrogel nanocomposites with structural features similar to native COL were fabricated. Interfacial mixing of oppositely charged polysaccharides gellan gum (GG), ALG, CS and kappa carrageenan (KCA) resulted in the fabrication of three different polyionic cross-linked hydrogels (CS-GG, CS-ALG, CS-KCA) with nanoscale structural features, similar to native COL, and pronounced osteogenic potential. Incubation of these fibrous hydrogels in simulated body fluid for three days promoted biomimetic apatite-like mineral deposition in vitro, showing a crystalline structure on the surface with amorphous calcium phosphate inside. CS-GG and CS-KCA exhibited a greater extent of mineralization compared to CS-ALG, as was confirmed by alizarin red staining. When the three different hydrogels were evaluated in a non-load bearing critical-size mouse calvarial defect model for 12 weeks, the CS-KCA non-mineral hydrogel significantly enhanced bone regeneration compared to empty defects. Moreover, the lyophilized form of the CS-KCA mineral hydrogel was more efficacious in regenerating mouse calvarial defects than the empty control as well as a COL sponge [111]. The better performance of CS-KCA was attributed to the sulfate group, which was postulated to improve bone regeneration through its greater ability to bind to proteins, including endogenously produced growth factors [132], preventing their denaturation, prolonging their efficacy [133,134,135] and resulting in stem cell and osteoblast recruitment from the surrounding areas. The success of these bioinspired hydrogels in enhancing bone regeneration without any added growth factors could be attributed to the influence of organized anisotropic assembly of mineral-fiber composites that facilitated protein adsorption, relying on the nano-/submicron-scale surface roughness and surface electrostatic charges [136,137].

Carboxymethyl chitosan (CMCS) is a water-soluble CS derivative, where the negatively charged carboxyl group binds to the positively charged CS to form NPs [138]. SDF-1α is a highly basic protein that plays a critical role in homing and localization of MSCs [139,140] and can bind electrostatically to the negatively charged CMCS. Thus, in situ-controlled release of SDF-1α could be an interesting strategy to effectively improve the osteogenic differentiation potential of recruited MSCs in bone regeneration. In an attempt to achieve this goal, SDF-1α/CS/CMCS NPs were prepared and characterized for various parameters including morphology, particle size, zeta potential, loading efficiency and the release characteristics from thermosensitive CS/β-glycerol phosphate disodium salt (GP) hydrogels [112,141]. The SDF-1α/CS/CMCS NPs within the CS/GP hydrogels showed a significantly sustained initial release of SDF-1α, 85% in the first four days and nearly 90% after 12 days, while, after 28 days, the cumulative release rate was much lower (only 40%). Additionally, the SDF-1α/CS/CMCS NP-embedded CS/GP hydrogel group significantly promoted new bone formation histologically, as compared to the SDF-1α-embedded hydrogels group and the empty control group, eight weeks following implantation in critical-size defects in rats [112]. In a further investigation, SDF-1α was directly added into a CS/GP hydrogel and anti-miRNA-138 was encapsulated by CS-based NPs and then embedded within the CS/GP hydrogel. Both the SDF-1α/hydrogel group and the SDF1α/NPs/hydrogel group cultured with bone marrow MSCs released SDF-1α and kept their bioactivity for six days. The dual release of SDF-1α and CS/anti-miRNA-138 NPs did not have synergistic effects on MSC migration compared with the SDF-1α/hydrogel group. Moreover, both the NPs/hydrogel and SDF-1α/NPs/hydrogel groups promoted the expression of osteogenesis-related genes, COL-1, osteopontin (OPN) and osteocalcin (OCN), at 3, 7, 14 and 21 days, without significant differences between them [113].

In a rat cranial critical-size defect model, an SDF-1α/NPs/hydrogel group demonstrated higher bone formation radiographically and histologically eight weeks post-surgically compared to the SDF-1α/hydrogel and NPs/hydrogel groups. Immunohistochemically, the SDF-1α/NPs/hydrogel group further demonstrated the strongest positive staining of COL-1, OPN and OCN in the tissues around the CS/GP hydrogel, indicating that the dual release of SDF-1α and CS/anti-miRNA-138 NPs might have significantly enhanced the expression of osteogenic proteins. It was further suggested that a fast release of SDF-1α from the hydrogel matrix combined with a slow release of anti-miRNA-138 from the NPs/hydrogel composite system could have provided in situ a precise control for endogenous cell homing and osteogenic differentiation of recruited MSCs with enhanced new bone formation [113].

Statins also have been suggested as potentially effective pro-regenerative drugs relying on their pleiotropic properties [142], with effects depending on the local concentration of the drug [143]. As atorvastatin and lovastatin are two lipophilic statins that are insoluble in an aqueous solution [144], a nano-emulsions-based drug delivery system has been synthesized using vitamin E acetate associated with D-α-Tocopherol polyethylene glycol succinate (TPGS) to encapsulate them in nanodroplets, in order to increase their aqueous solubility and to improve their bioavailability. The thermosensitive CS hydrogel functionalized by atorvastatin TPGS nano-emulsions or lovastatin TPGS nano-emulsions was examined in a mice calvarial bone defect model, demonstrating a significant increase in neo-bone formation histomorphometrically two weeks post-surgically. Moreover, the soft tissue surrounding the functionalized hydrogels showed a significant reduction in inflammatory cellular infiltration [114].

##### Chitosan and Dextran-Based Nanohydrogels

Nanocomposite scaffolds based on hydroxypropyl chitosan/aldehyde dextran (CD) hydrogels and strontium-nanohydroxyapatite (Sr-nHAp) NPs were fabricated at ratios of Sr/(Sr + Ca) of 0% (nHAp), 50% (Sr50nHAp) and 100% (Sr100nHAp) incorporated into the CD hydrogel. The results showed that either nHAp or Sr-nHAp NP incorporation into CD hydrogel significantly improved the rheological and mechanical properties. The Sr^2+^ released from the Sr100nHAp/CD hydrogel was in the range of optimal concentration for pro-osteogenesis. The enhanced alkaline phosphatase (ALP) activity, OCN secretion and cell mineralization of the Sr-nHAp/CD hydrogel indicated that the incorporation of Sr-nHAp promoted MC3T3-E1 pre-osteoblast cell line differentiation at both early and late differentiation stages. The osteogenic ability of the Sr100nHAp/CD hydrogel became more significant over time, as deduced from the upregulated expression of runt-related transcription factor 2 (RUNX2), ALP, OCN and COL-1 by using Sr100nHAp/CD at 14 days. The Sr^2+^ modification exerted an obvious effect on the positive transformation of inflammation-related macrophages (M1) to healing-related macrophages (M2) phenotype gene expression. The Sr100nHAp/CD hydrogel showed the highest promotion on the polarization of macrophages towards the M2 phenotype, as was indicated by lower expression levels of tumor necrosis factor (TNF-α) and higher levels of interleukin (IL)-10, as compared to the CD hydrogel group. The M2-activated macrophages strongly promoted homing and osteogenic differentiation of MC3T3-E1 on the hydrogels by upregulating osteogenic cytokines including transforming growth factor β (TGF-β) and BMP-2 [115].

##### Hyaluronic Acid-Based Nanohydrogels

D-glucuronic acid and *N*-acetyl-D-glucosamine combine to form the anionic polysaccharide known as hyaluronic acid (HA) [145]. It is highly hydrophilic with a moisturizing effect, owing to the hydroxyl group that can bind water molecules tightly to the chain via hydrogen bonds [146]. Because of their high biocompatible characteristics, HA-based NGs have demonstrated considerable promising attributes in nanotherapeutics and nanomedicine [145] as well as drug delivery systems [147,148,149].

Injectable biomaterials that are non-immunogenic and degrade in a controlled fashion, such as HA, with enhanced carrier properties were developed in order to increase the efficacy of BMP-2 in bone treatment [150,151]. Consequently, HA-based hydrogels containing HAp NPs (HAp yielded cohesive and viscous pastes, improving the physical properties of the gels) and different concentrations of BMP-2 (0, 5, and 150 mg/mL) were tested in mandibular bone augmentation approaches in rats. Subperiosteal injections with the scaffolds resulted in mandibular augmentation, with an increase in bone volume percentage histologically, correlating with the amount of BMP-2 within the hydrogel formula (0, 5 and 150 mg/mL resulting in ~8%, ~14% and ~58% bone augmentation, respectively). In addition, immunohistochemical characterization revealed high expressions of OCN, OPN and COL-IV with no fibrous encapsulation [116].

##### Chitosan and Hyaluronic Acid-Based Nanohydrogels

In an attempt to overcome the previously-mentioned limitations of HAp, it was combined with polysaccharide-based hydrogels (with their advantageous properties of self-repair, being injectable, excellent biocompatibility and biodegradability) [152] and a composite hydrogel consisting of *N*-carboxyethyl CS-HA-aldehyde/nHAp was synthesized. As compared to the control and hydrogel groups, the hydrogel/nHAp group promoted osteogenic differentiation of MC3T3-E1 cells in vitro, which was confirmed by a significant increase in ALP activity and alizarin red positive bone-like inorganic calcium deposits. The osteopromotive effect of the hydrogel/nHAp group scaffold was further evaluated in vivo using a rat alveolar bone extraction model in the mandibular central incisor. After four weeks, 3D reconstruction of new bone tissue in the entire alveolar fossa was observed in the hydrogel/nHAp group as revealed by micro-computed tomography (µ-CT) and histologically [117]. This was primarily attributed to the dissolving nHAp crystals with their ability to stimulate stem cell recruitment for bone regeneration [153,154]. The absence of significant inflammation or immune response after hydrogel/nHAp injection might additionally have provided a suitable environment for osteoblast proliferation and differentiation [117].

##### Collagen and Alginate-Based Nanohydrogels

COL and AG can work together to combine their positive qualities and get over each material’s drawbacks. The weak mechanical properties of COL and the inherent lack of cell-binding motifs inside AG, which are the main hurdles that restrict their wide range of application, can be overcome by their combination with upregulated cell-binding motifs and enhanced mechanical properties. Moreover, the ease of gelation of this composite under mild conditions enables the retention of bioactive agents and enhances cell encapsulation [155].

Nerve growth factors (NGF) proved to play an important role in bone regeneration [156,157]. One of the limitations of human NGF is its rapid clearance from the body by enzymatic degradation. Thus, to overcome this limitation, NGF has been incorporated into COL/nHAp/AG hydrogel and investigated for its bone formation potential in a rabbit mandibular distraction osteogenesis model (0.75 mm/12 h for six days). The rabbits were divided into four groups, group I received injections of COL/nHAp/AG hydrogel containing human NGFβ, groups II, III and IV received injections of human NGFβ, COL/nHAp/AG hydrogel and saline, respectively. After 14 days, no difference in bone dimensions was observed among the four groups. On the contrary, bone mineral density (BMD), maximum loading and the bone volume/total volume (BV/TV) of the new bone in the distraction gap were significantly greater in group I than in the other three groups. The findings proved the osteoconductive activity of the hydrogel system and that human NGFβ was able to retain its biological activities for a prolonged period until its release from the NPs/hydrogel system [118].

#### 3.1.2. Synthetic Polymer-Based Nanohydrogels (Table 2) (Figure 5)

Table 2 and Figure 5 are shown below.

**Figure 5 biomolecules-13-00205-f005:**
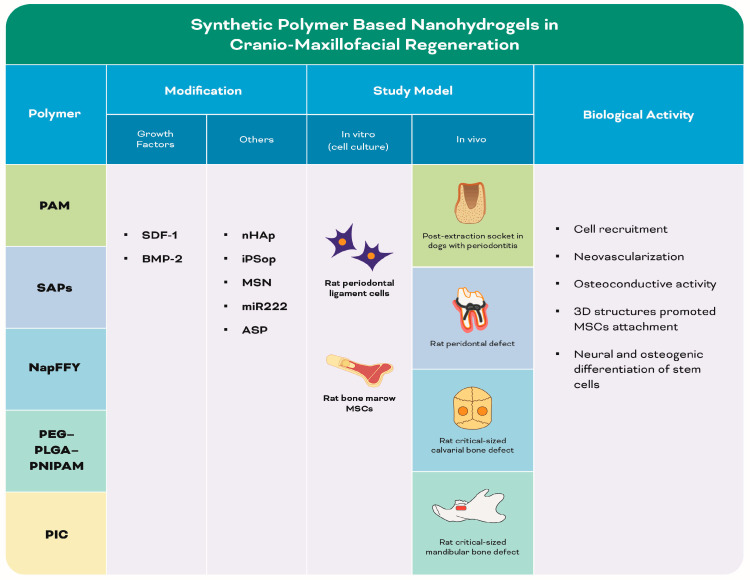
Infographic illustrating experimental studies of synthetic polymer-based nanohydrogels in Table 2.

**Table 2 biomolecules-13-00205-t002:** Summary of the included studies employing different synthetic polymer-based nanohydrogels in cranio-maxillofacial regeneration.

Author, Year	Polymer	Co-Polymer	Modification	Main Features of Nano-Polymer	Study Model		Biological Activity	Outcomes
					In Vitro (Cell Culture)	In Vivo		
Tanongpitchayes et al., 2021 [158]	PAM		nHAp			Post-extraction socket in dogs with periodontitis	Hydrogel promoted cell infiltration and neovascularization	nHAp-based hydrogel enhanced alveolar bone regeneration after 12 weeks
Takeuchi et al., 2016 [159]	SAPs (RADA16)			Nanofibres with nanopores	Rat periodontal ligament cells	Rat peridontal defect	Nanostructure facilitated cell recruitment and angiogenesis.	RADA16 hydrogels enhanced periodontal defect healing after 4 weeks
Hayashi et al., 2016 [160]	SAPs		iPSop	Nanofibers		Rat critical-sized calvarial bone defect	-Secreted growth factors and cytokines -Enhanced the osteoconductivity of thehydrogel	iPSop encapsulated in SAPs nanofiber hydrogel induced bone regeneration after 4 weeks
Tan et al., 2019 [161]	NapFFY		SDF-1, BMP-2	Nanofibers	Rat bone marrow MSCs	Rat critical-sized periodontal defect	-SDF-1 recruited MSCs to the defect site, while differentiation was promoted by BMP-2 -3D nanofiber structures of the hydrogel promoted MSC attachment	SDF-1/BMP-2/NapFFY hydrogel promoted periodontal bone regeneration after 8 weeks
Lei et al., 2019 [162]	PEG–PLGA–PNIPAM		MSN, miR222, ASP	Microspheres		Rat critical-sized mandibular bone defect	miR222 induced neural differentiation of stem cells. ASP induced a pro-osteogenic microenvironment at defect sites	miR222/MSN/ASP hydrogel induced innervated bone tissue formation after 10 weeks
Cui et al., 2019 [163]	PIC	CNTs		3D scaffold with interconnected grid structure	Rat bone marrow MSCs	Rat critical-sized calvarial bone defect	CNTs into the PIC hydrogels promoted neovascularization and osteogenesis	PIC/MWCNT scaffolds enhanced bone repair after 8 weeks

Polyacrylamide: PAM; nanohydroxyapatite: nHAp; self-assembling peptides: SAPs; Arginine-alanine-aspartate-alanine: RADA; human induced pluripotent stem cell-derived osteoprogenitors: iPSop; Nap-Phe-Phe-Tyr-OH: NapFFY; stromal cell-derived factor-1: SDF-1; bone morphogenetic protein: BMP; mesenchymal stem cells: MSCs; three-dimensional: 3D; poly(ethylene glycol)-b-poly(lactic-co-glycolic acid)-b-poly(*N*-isopropylacrylamide): PEG–PLGA–PNIPAM; mesoporous silica nanoparticle: MSN; microRNA: miR; aspirin: ASP; polyion complex: PIC; carbon nanotubes: CNTs; multiwalled carbon nanotubes: MWCNT.

##### Polyacrylamide-Based Nanohydrogels

The effectiveness of nHAp-based hydrogels made up of PAM with nHAp was evaluated in the preservation of post-extraction sockets in a dog periodontitis model. Results demonstrated that the radiographic grading, bone height measurement and bone regeneration analysis were positively significant at all follow-up times (two, four, eight, and twelve weeks post-operation) compared to baseline. Moreover, scanning electron microscopy (SEM) with energy-dispersive X-ray spectroscopy imaging after immersion in simulated body fluid for 14 days showed a homogeneous distribution of new apatite formation on the hydrogel surface. This proved the osteoconductive ability of the nHAp/PAM hydrogel in promoting cell infiltration and neovascularization in the alveolar bone regeneration process [158].

##### Self-Assembling Peptide-Based Nanohydrogels

Self-assembling peptides (SAPs) derived from essential amino acids have similar biological properties to the native ECM [164]. SAPs have received attention as scaffolds in tissue regeneration due to their cell adhesive properties and biocompatibility [165,166,167], providing a 3D microenvironment with specific properties that facilitate in vitro proliferation and migration of various cell types [168,169]. The SAPs hydrogel is not only biocompatible but also has the potential to be modified at the molecular level. In vitro, periodontal ligament cells grown on RADA16 (arginine-alanine-aspartate-alanine) showed a gradual increase in proliferation up to 72 h. In a rat model with extracted maxillary first molars and surgically-created bilaterally mesial periodontal defects in second molars, the defects treated with RADA16 revealed significantly greater bone volume fraction and trabecular thickness than those treated with Matrigel or left unfilled after four weeks. Histologically, enhanced new bone formation was observed in the RADA16 group. Additionally, periodontal-like collagen bundles ran obliquely to the root surface in the RADA16 group. Expression levels of proliferating cell nuclear antigen-positive cells, vascular endothelial growth factor (VEGF) and OPN in the RADA16 group were significantly greater than those in other groups [159].

Human induced pluripotent stem cell (iPSC)-derived osteoprogenitor (iPSop) cells were encapsulated on SAP nanofiber scaffolds. Rat critical-sized calvarial bone defects were assigned to test the healing effects of the nanofiber scaffold alone (nanofiber), nanofiber scaffold with iPSop cells (nanofiber + iPSop) or a physiological salt solution (saline) as a control [160]. Compared with the saline and nanofiber groups, the nanofiber + iPSop group had better regeneration with significantly higher bone volume, as revealed radiographically and histologically. Medullary cavities with numerous capillaries were present in the regenerated bone tissues of the nanofiber + iPSop group, indicating mature bone tissue. The better regeneration and vascularization of the nanofiber + iPSop than the SAP nanofiber hydrogel alone could be attributed to the growth factors and cytokines secreted by transplanted osteoprogenitors derived from human iPSCs [160].

Supramolecular hydrogels, formed by the self-assembly of small molecules through non-covalent interactions [170], have been further introduced as a solution to overcome the drawbacks of synthetic polymers, which limit their clinical translation including less biocompatibility and biodegradability [171,172]. A biocompatible supramolecular hydrogel Nap-Phe-Phe-Tyr-OH (NapFFY) was synthesized to encapsulate SDF-1 and BMP-2 (at an optimum concentration of 500 μg/L for each of them). In vitro and in vivo results indicated that these two bioactive factors were ideally, synchronously and sustainably released from the hydrogel to effectively promote the regeneration and reconstruction of periodontal bone tissues. The quantitative polymerase chain reaction results of ALP mRNA expression revealed that bone marrow MSCs were recruited to the defect sites by SDF-1 and their differentiation was promoted by BMP-2 released from the NapFFY hydrogels. After the bone defect areas were treated with SDF-1/BMP-2/NapFFY hydrogel for eight weeks in a maxillary critical-sized periodontal bone defect rat model, a superior bone regeneration rate of 56.7% bone volume fraction was achieved, as compared with 34.9% for the SDF-1/NapFFY hydrogel or 36.6% for the BMP-2/NapFFY hydrogel [161].

##### Polyethylene Glycol-b-poly(lactic-co-glycolicacid)-b-poly(*N*-isopropylacrylamide)-Based Nanohydrogels

Mesoporous silica (MS) NPs have been used as a nanocarrier owing to their numerous advantages, which include their large superficial area, abundant pore size, well-demarcated pore structure, excellent biocompatibility and a surface that is highly amenable to functionalization [173]. An injectable thermos-responsive MSNPs-embedded core-shell structured PEG-b-poly(lactic-co-glycolic acid) (PLGA)-b-poly(*N*-isopropyl acrylamide) hydrogel was fabricated for localized and long-term co-delivery of miR222 and aspirin (ASP) in the form of an miR222/MSNPs/ASP [162]. A rat critical-size mandibular defect model was used to examine the capacity of the miR222/MSNPs/ASP hydrogel for neurogenic induction during bone formation. The animals were divided into MSNPs, MSNPs/ASP and miR222/MSNPs/ASP hydrogel groups. After 10 weeks, µ-CT and histological evaluations of the harvested mandible bone tissue specimens revealed a significantly higher percentage of new BV/TV in the miR222/MSNPs/ASP group than in the other groups. A comparison of the MSNPs and MSNPs/ASP groups revealed no significant difference in new bone formation, indicating that ASP alone could not promote osteogenesis in a rat mandibular defect. Fluorescent immunohistochemistry demonstrated that the neural-related protein Tuj1 and glial protein S100 were further highly expressed in the newly formed bone tissue of the miR222/MSNPs/ASP group. Moreover, immune expression of calcitonin gene-related peptide (CGRP) was observed in the miR222/MSNPs/ASP group but not in the MSNPs/ASP group, underlying the role of miR222 in enhancing neurogenic differentiation of bone marrow MSCs into neural-like cells secreting CGRP. Subsequently, ASP further increased the osteogenic potential in CGRP-stimulated bone marrow MSCs near neural-like cells, thus achieving accelerated bone regeneration with better innervation [162].

##### Polyion and Carbon Nanotube-Based Nanohydrogels

A tough polyion complex (PIC) hydrogel was synthesized, and multiwalled carbon nanotubes (MWCNTs) were incorporated into the PIC matrix to form the PIC/MWCNT biohybrid hydrogel, which was manufactured into 3D scaffolds by extrusion-based 3D printing. In vitro, rat bone marrow MSCs demonstrated enhanced osteogenic differentiation as well as increased biocompatibility with the PIC/MWCNT scaffold than with the PIC scaffold [163]. An in vivo experiment using a rat calvarial bone defect revealed a significant increase in BV/TV ratio and the values of BMD in the PIC/MWCNT group, larger than for the PIC group after two to eight weeks as demonstrated by µ-CT and histological analysis. Additionally, immunological staining demonstrated that the OCN, RUNX2, COL-1 and CD31 were expressed abundantly in the PIC/MWCNT hydrogel group. The study concluded that this functionalized composite hydrogel (PIC/MWCNT) combined the osteogenic effects of CNTs and the suitable mechanical properties of PIC hydrogels, opening new perspectives for bone TE [163].

#### 3.1.3. Natural and Synthetic Polymer (Composite)-Based Nanohydrogels (Table 3) (Figure 6)

Table 3 and Figure 6 are shown below.

**Figure 6 biomolecules-13-00205-f006:**
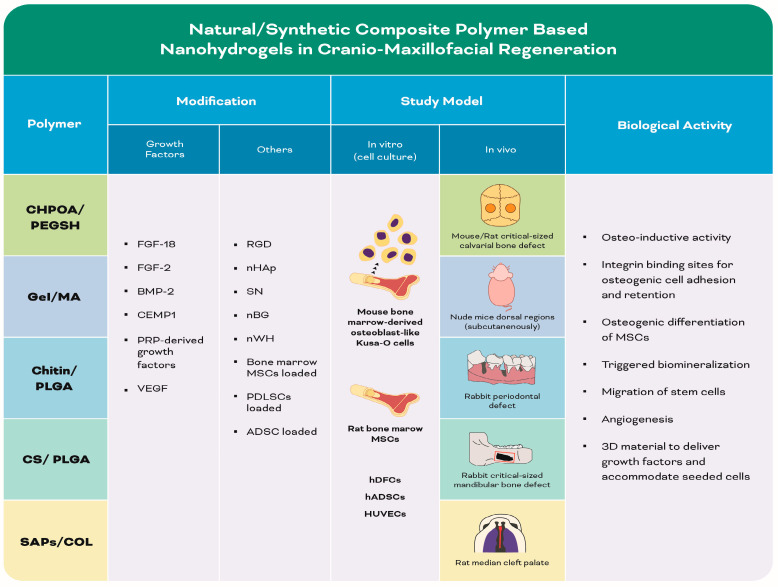
Infographic illustrating experimental studies of natural and synthetic polymer (composite)-based nanohydrogels in Table 3.

**Table 3 biomolecules-13-00205-t003:** Summary of the included studies employing different natural/synthetic composite polymer-based nanohydrogels in cranio-maxillofacial regeneration.

Author, Year	Polymer	Co-Polymer	Modification	Main Features of Nano-Polymer	Study Model		Biological Activity	Outcomes
					In Vitro (Cell Culture)	In Vivo		
Fujioka-Kobayashi et al., 2012 [174]	CHPOA	PEGSH	FGF-18, BMP-2	Nanogel		Mouse critical-sized calvarial bone defect	-Sustained release of FGF-18 enhanced the osteo-inductive activity of BMP-2 by downregulation of BMP antagonist (Noggin)	CHPOA- PEGSH/FGF-18 + BMP-2 hydrogel induced effective bone repair after 8 weeks
Charoenlarp et al., 2018 [175]	CHPOA	PEGSH	FGF-18, BMP-2, RGD	Nanogel	Mouse bone marrow-derived osteoblast-like Kusa-O cells	Mouse critical-sized calvarial bone defect	The initial release of growth factors from scaffold recruited osteoprogenitor cells to the defect site and then RGD peptides provided integrin binding sites on the surface of the material for osteogenic cell adhesion and retention	RGD-NanoCliP disc with growth factors showed a significant acceleration of bone healing after 8 weeks.
Shi et al., 2021 [176]	Gel	MA	Rat bone marrow MSCS, nHAp, SN	Interconnected porousnetwork		Rat critical-sized calvarial bone defect	-nHAp similar to natural bone, preserving the cellular bioactivity of the encapsulated MSCs -SN induced osteogenic differentiation of MSCs	MSCs-loaded GelMA-nHAp-SN hydrogels stimulated bone regeneration after 8 weeks.
Chen et al., 2016 [177]	Gel	MA	hPDLSCs, nHAp	3D interconnected porous structure		Nude mice dorsal regions (subcutenously)	nHAp enhanced surface topographical properties, which promoted cell adhesion	hPDlSCs-laden GelMA/nHAp microgels enhanced new bone formation after 8 weeks
Sowmya et al., 2017 [178]	Chitin	PLGA	nBG, rhCEMP1, rhFGF-2, PRP-derived growth factor	Tri-layered porous scaffold	hDFCs	Rabbit periodontal defect	-nBG triggered biomineralization -Growth factors facilitated migration and differentiation of stem cells	Chitin–PLGA/nBG/CEMP1), chitin–PLGA/FGF-2 and chitin–PLGA/nBG/PRP layers induced a complete defect closure and periodontal regeneration after 3 month
Amirthalingam et al., 2021 [179]	Chitin	PLGA	nBG, nWH, FGF-18		hADSCs, HUVECs	Mice critical-sized calvarial bone defect	Mg2+ improved proangiogenic and osteogenic properties of nWH -Si^4+^ in nBG enhanced angiogenesis - FGF-18 osteogenic differentiation role was enhanced	Chitin–PLGA/nWH-FGF significantly promoted bone regeneration after 8 weeks
Wang et al., 2020 [180]	CS	PLGA	BMP-2, VEGF, ADSC, nHAp	3D porous structure		Rabbit critical-sized mandibular bone defect	-CS, nHAp and PLGA microspheres generated a 3D material to deliver growth factors and accommodate seeded cells -BMP-2 and VEGF promoted angiogenesis and osteogenesis	BMP-2/VEGF-loaded injectable nHAp/PLGA/CS hydrogel promoted bone formation after 12 weeks
Mostafa et al., 2015 [181]	SAPs (RADA4)	COL	BMP-2	Nanofibers		Rat median cleft palate	BMP-2 induced oseoinductivity	Hydrogel/BMP-2 enhanced new bone formation after 8 weeks

Cholesteryl group- and acryloyl group-bearing pullulan: CHPOA; thiol-bearing polyethylene glycol: PEGSH; fibroblast growth factor: FGF; bone morphogenetic protein: BMP; arginine-glycine-aspartate: RGD; gelatin: Gel; methacryloyl: MA; mesenchymal stem cells: MSCs; nanohydroxyapatite: nHAp; nanosilicate: SN; human periodontal ligament stem cells: hPDLSCs; three-dimensional: 3D; poly lactic-co-glycolic acid: PLGA; nano-bioactive glass: nBG; recombinant human: rh; cementum protein: CEMP; platelet-rich plasma: PRP; human dental follicle stem cells: hDFCs; whitlockite nanoparticles: nWH; human adipose-derived mesenchymal stem cells: hADSCs; human umbilical vein-derived endothelial cells: HUVECs; chitosan: CS; vascular endothelial growth factor: VEGF; self-assembling peptides: SAPs; arginine-alanine-aspartate-alanine: RADA; collagen: COL.

##### Pullulan and Polyethylene Glycol-Based Nanohydrogels

Maltotriose units make up the polysaccharide pullulan, sometimes referred to as 1,4- and 1,6-glucan [182,183]. Hydrophobes such as cholesterol modify the pullulan polymer, causing it to behave as amphiphilic molecules that could serve as superior NG carriers with amphiphilic characteristics [184]. Due to the variations in functional derivatives that cause pullulan to modify its characteristics, pullulan has been extensively used in NGs for bone TE [72].

In this context, cholesteryl group- and acryloyl group-bearing pullulan (CHPOA) NGs were aggregated to form fast-degradable hydrogels (NanoClik) by cross-linking with thiol-bearing PEG. CHPOA NGs containing recombinant human FGF-18, BMP-2 or combined were prepared. The NanoClik-FGF-18 + BMP-2/hydrogel treatment strongly enhanced and stabilized the BMP-2-dependent bone repair in a critical-size calvarial defect, inducing osteoprogenitor cell infiltration inside and around the hydrogel [174]. This was primarily attributed to the ability of FGF18 to enhance the osteoinductive activity of low-dose BMP-2 by downregulation of BMP antagonist Noggin [185].

To prepare different physical forms of the NGs, the NanoClik hydrogel was conventionally dried to prepare disc-shaped NG-cross-linked gel (NanoClik disc), or freeze-dried to prepare NG-cross-linked porous (NanoCliP) discs. Additional modification was performed by adding arginine-glycine-aspartic acid (RGD) peptides to prepare RGD-NanoCliP discs. The different forms of the NGs were used as carriers for growth factors in different clinical applications. µ-CT results revealed that the NanoClik disc and NanoCliP disc containing growth factors (human FGF-18 + human BMP-2) induced osteogenic closure of bone defects at 59.2% and 65.6%, respectively, with no significant difference. RGD-NanoCliP discs containing phosphate buffer saline or growth factors showed a significant acceleration of bone healing (28.3% and 83.9%) after eight weeks, respectively [175]. In comparison to the freshly made NanoClik hydrogel with incorporated human BMP-2 and human FGF18, which induced almost perfect closure in 90% of the defects, with an average healing rate of 93.4% after eight weeks [174], both NanoClik and NanoCliP discs with growth factors failed to reach similar healing potential [175]. Moreover, the RGD-NanoCliP disc revealed better attachment of mouse bone marrow-derived osteoblast-like Kusa-O cells, with a significantly higher number of cells than the NanoCliP disc. Histological results revealed that RGD-NanoCliP discs containing growth factors induced thicker, more extensive and more calcified trabecular bone formation than NanoCliP discs containing growth factors [175], yet less than the NanoClik hydrogels with growth factors. This was attributed to changes that occurred in the gel texture, thickness, stiffness and shrinkage due to the drying technique that resulted in difficulty in the adaptation of the gel margin to the defect. However, the reduction in bone regeneration due to the drying of the hydrogel by conventional or freeze-drying method was overcome by the addition of RGD peptides in the system [175].

##### Gelatin and Methacryloyl-Based Nanohydrogels

A study designed a MSCs-laden, nHAp and nanosilicate (SN)-loaded bone mimetic and injectable Gel and methacryloyl (MA) GelMA-nHAp-SN hydrogel system for bone TE. Introducing HAp in GelMA provided a compositional similarity to the natural bone ECM, while SN provided ideal injectability and osteoinductivity. The GelMA-nHAp-SN nanocomposite hydrogel presented the highest cellular viability with the most sizable cell-spreading area as compared to the GelMA-nHAp and GelMA-SN hydrogels [176]. GelMA/nHAp microgel arrays were fabricated by blending different weights of GelMA solution (5% and 10% *w*/*v*) with nHAp of varying concentrations (1%, 2% and 3% *w*/*v*). SEM images indicated that the pore size of hydrogel decreased with increasing nHAp fraction, while the wall surface of micropores became rougher. The stiffness of microgels was enhanced by increasing both monomer and nHAp concentration, while the swelling ratio of GelMA/nHAp microgels was only related to GelMA monomer fraction [177].

GelMA (10% *w*/*v*) microgel was selected for further studies, owing to its remarkable mechanical properties, which meet the mechanical properties of neonatal alveolar bone tissues. Interestingly, the viability of human PDLSCs in the 10% *w*/*v* GelMA groups after 24 h encapsulation was not affected when nHAp concentration was 1% w/v and 2% *w*/*v*, while the number of dead cells increased with the 3% *w*/*v* nHAp group. The BrdU assay also revealed similar results. Additionally, the human PDLSCs encapsulated in 2% nHAp with 10% *w*/*v* GelMA microgels exhibited significant mineralization after 10 days and significantly expressed the osteogenic differentiation genes, including ALP, bone sialoprotein (BSP), OCN and RUNX2 after 14 days of culture compared to microgels with other ratios. Ectopic transplantation subcutaneously into the backs of nude mice showed that GelMA/nHAp microgels (10%/2% *w*/*v*) increased mineralized tissue formation with abundant vascularization, compared with the 1%, 3% and the pure GelMA group. On the contrary, Masson’s histological staining results indicated that a certain amount of periodontal ligament-like tissue and blood vessels formed in the control, 1% and 2% nHAp groups [177].

##### Chitin/Chitosan and poly(lactic-co-glycolic acid)-Based Nanohydrogels

Tri-layered nanocomposite hydrogel scaffold was developed by assembling CN-PLGA/nanobioactive glass (nBG) ceramic/cementum protein 1 (CEMP1) (CN-PLGA/nBG/CEMP1) for cementum regeneration, CN-PLGA/FGF2 for periodontal ligament regeneration and CN-PLGA/nBG/platelet-rich plasma-derived growth factors for alveolar bone regeneration [178] to imitate the complex periodontal hierarchical architecture [186]. The presence of specific growth factors and/or recombinant proteins in the scaffolds contributed to the enhancement of human dental follicle stem cells (DFSCs) adhesion and proliferation. The cementogenic differentiation, assessed by cementogenic proteins (COL-1, CEMP1 and BSP) expressions, the fibrogenic differentiation, assessed by fibrogenic proteins (fibroblast surface protein, COL-1 and periodontal ligament associated protein 1 (PLAP1)) expressions and the osteogenic differentiation, assessed by the osteogenic proteins (RUNX2, COL-1 and OCN) expressions by human DFSCs on the tri-layered nanocomposite hydrogel scaffold with growth factors were comparable to the cellular differentiation potentials on the tri-layered nanocomposite hydrogel scaffold cultured in induction medium [178]. In vivo, the tri-layered nanocomposite hydrogel scaffold with/without growth factors was implanted into rabbit maxillary periodontal defects and compared with the controls at one and three months postoperatively. The tri-layered nanocomposite hydrogel scaffold with growth factors demonstrated complete defect closure and healing with new cancellous-like tissue formation on µ-CT analysis. Histological and immunohistochemical analyses for CEMP1, PLAP1, COL-1 and OCN further confirmed the formation of new cementum, fibrous periodontal ligament and alveolar bone with well-defined bony trabeculae upon using tri-layered nanocomposite hydrogel scaffold with growth factors in comparison to the control groups (sham and positive) and the tri-layered nanocomposite hydrogel scaffold without growth factors group [178].

As mentioned previously, HAp is the major bone mineral present in the body, mainly in a carbonated form containing trace elements such as Mg^2+^, sodium, Si^4+^ and Zn^2+^ that are considered essential in bone metabolism [187]. Si^4+^ and Mg^2+^ play an important role in enhancing new bone and blood vessel formation. Among the Mg^2+^-containing bio-ceramics, whitlockite (WH) is the second most abundant bone mineral present in the human body, which occupies about 25 wt% and 26–58 wt% in bone [188,189]. WH NPs (nWH) demonstrated superior osteogenic potential and inhibition of osteoclastogenesis compared to nHAp. Increased bone regeneration potential could be ascribed to the release of Mg^2+^ ions, which was found to increase osteogenic and angiogenic marker expression [190,191]. In addition, the altered release of Ca^2+^ and phosphate ions from nWH as compared to nHAp would have also played a role in enhancing bone regeneration. Si^4+^ containing BG was found to possess biocompatibility, osteoconductivity and osteostimulation [192].

An injectable CN-PLGA hydrogel containing nBG 10% *w*/*w*, or nWH 5% w/w with FGF-18, was investigated regarding the osteogenic and neo-bone formation potential against commercially available nHAp 5% w/w with FGF-18 loaded on CN-PLGA hydrogel in a critical-sized defect region [179]. In vitro studies using human adipose-derived MSCs (ADSCs) and human umbilical vein-derived endothelial cells revealed that the CN-PLGA nWHF (nWH + FGF-18 containing CN-PLGA) group had the highest osteogenic potential, ALP activity, BMP-2 quantification and osteogenic gene expressions for RUNX2, ALP, COL-1A and OCN at day 7 and 14. The tube formation assay demonstrated that nWH and nBG-containing hydrogels stimulated the formation of tubular structures [179]. The presence of Mg^2+^ ions induced tubular-like structures by increasing the expression of VEGF-A and nitric oxide synthase [193,194] in the CN-PLGA nWH and CN-PLGA nWHF hydrogel groups. Furthermore, Si^4+^ ions were found to promote angiogenesis in the CN-PLGA nBG and CN-PLGA nBGF (nBG + FGF-18 containing CN-PLGA) groups. On the contrary, nHAp-containing hydrogel systems with or without FGF-18 did not induce any tubular-like structures. In vivo bone regeneration studies displayed near-complete bone regeneration for CN-PLGA nWHF, where its BV/TV% was the highest (synergistic effect) compared to CN-PLGA nBGF and nHAp with FGF-18 (additive effect) eight weeks following implantation [179].

Furthermore, a thermosensitive hydrogel was prepared by integrating BMP-2/VEGF-loaded PLGA microspheres with nHAp and CS to accommodate ADSCs. The efficacy of this scaffold material for bone TE was investigated in a rabbit 8 mm-sized full-thickness mandibular bone defect model. Along different examination periods (four, eight and twelve weeks postoperatively) µ-CT imaging and histological observations consistently demonstrated synchronism of new blood vessel and bone formation, which was most significant in the BMP-2/VEGF-loaded ADSCs scaffold composite group and less in the group with BMP-2 alone. However, this effect was rather diminished in VEGF alone group and absent in the control group. It was thus suggested that the integration of CS, nHAp and PLGA microspheres generated a 3D material efficient to deliver growth factors and enhance the microenvironment for seeded cells. Moreover, the porous structure was beneficial for the transportation of nutrients and waste products, degradation of materials and ingrowth of blood vessels [180].

##### Collagen and Self-Assembled Peptide-Based Nanohydrogels

SAP nanofiber-based hydrogel RADA4, with an absorbable collagen sponge (ACS) backbone, was developed to deliver BMP-2 and control its release. ACS was utilized to provide structural integrity for the hydrogel, facilitate handling and prevent nanofiber migration away. Upon correlating the timeline of the bone healing cascade with the BMP-2 release data, the constructs containing 2% nanofiber were found to release minimal BMP-2 concentrations during the inflammatory phase and maximal concentrations during the cell recruitment phase. Subsequently, a scaffold with the appropriate nanofiber density (2%) was implanted into a rodent model of a cleft palate and bone healing was assessed using µ-CT and histology. The rats were assigned into control (no scaffold), ACS alone, ACS + BMP-2, nanofiber + ACS and nanofiber + ACS + BMP-2 groups [181]. The bone filling percentage in the nanofiber + ACS + BMP-2 group was significantly higher than other groups at weeks four and eight, however, this increase was not significant as compared to the ACS + BMP-2 group. Furthermore, significant bone bridging across the defect as early as four weeks was evident only in the nanofiber + ACS + BMP-2 group. Histological assessments of bony defects at week eight displayed fibrous tissue filling the defect, with limited bone regrowth at the defect margins in the ACS and nanofiber + ACS groups. On the other hand, ACS+BMP-2 treatment resulted in partial closure of the defect, while nanofiber + ACS + BMP-2 showed central and peripheral bone formation in the defect site. More mature bone and increased bone thickness in the regenerated area were observed in the nanofiber + ACS + BMP-2 group. These results suggested that the utilization of nanofiber hydrogel scaffold sustained the BMP-2 release, resulting in improved bone healing [181].

## 4. Conclusions and Future Perspectives

The insertion of nano/microstructures in hydrogel formulas aided in the creation of hybrid hydrogels with a variety of functions for use in biological systems. Particle incorporation and domain construction enable not only stimuli-responsive material behavior, tunable cellular response and targeted medication therapy, but also enhance mechanical and physical qualities. The favorable polymer nanocomposites characteristics, including the upregulated potential to enhance cell adhesion, proliferation, differentiation and formation of new bone tissue, make them promising candidates in the field of bone TE. As the development of nanohydrogels progresses, designing new systems that closely mirror the environment will continue. Nanohydrogels that can be precisely tailored in a modular manner for the desired application and those that can be easily manufactured to attain a high level of architecture in treating craniofacial defects remain to be the current choice in the TE field. Future research should focus on the co-administration of different molecules to create materials, using simple quick and affordable techniques, to simulate the natural microenvironment.

The list of all the abbreviations is available in Abbreviations.

## Figures and Tables

**Figure 1 biomolecules-13-00205-f001:**
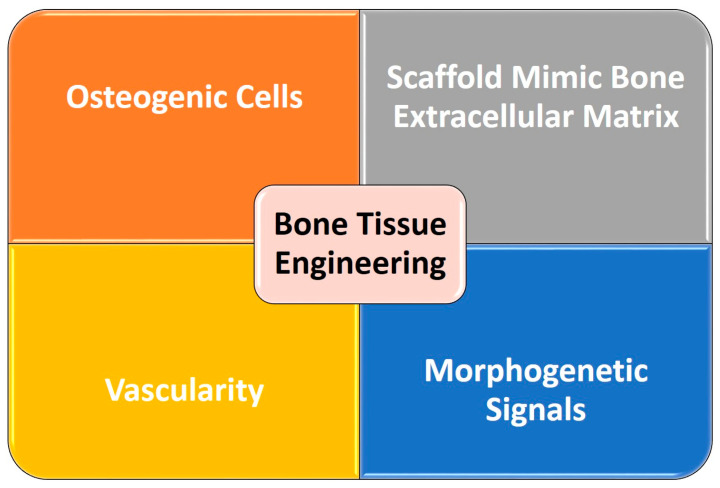
Diagram showing the key components in the field of bone tissue engineering.

**Figure 2 biomolecules-13-00205-f002:**
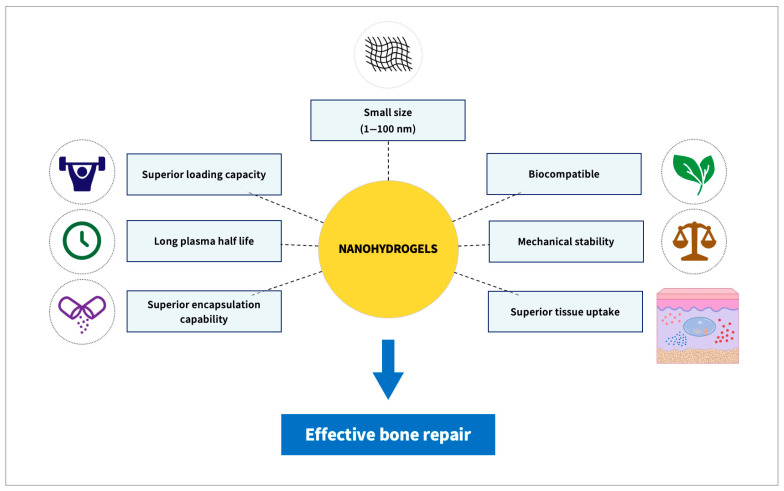
Diagram showing general characteristics of nanohydrogels.

**Figure 3 biomolecules-13-00205-f003:**
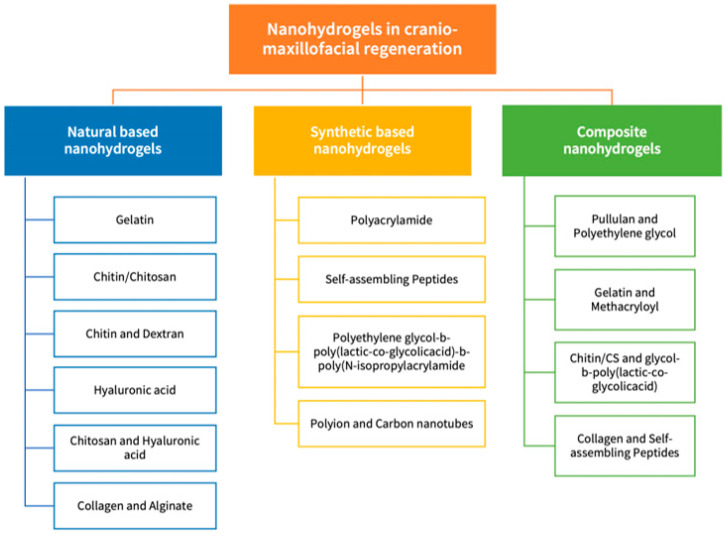
Diagram showing classifications of nanohydrogels employed in cranio-maxillofacial regeneration.

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
