# Peer review of "Polymeric Nanocomposite Hydrogel Scaffolds in Craniofacial Bone Regeneration: A Comprehensive Review"

_biomolecules, 2023, doi:10.3390/biom13020205_

Round 1

Reviewer 1 Report

The manuscript "Nano-based-polymeric hydrogels in craniofacial bone regeneration: A comprehensive review" presents a review about hydrogels for bone regeneration. This starts with the preparation and the use of different polymers (and possibly fillers), and ends with the analysis and review about the performance and use. So, this article is highly interesting for researchers working on man-made tissues for various reasons but also specifically for bone regeneration. The clear difference and the specifity of bone repair tissues did not become very clear to me. So this is a point to be addressed clearer for the broad readership. The whole manuscript is written well and considers much literature. So I would plead for publication after minor changes.

So for instance there are tissues for drug delivery (https://doi.org/10.1021/acs.macromol.1c01596). What is the crucial difference between these and bone repair tissues?

Author Response

Thank you for the comprehensive review of our manuscript entitled " Nano-based-polymeric hydrogels in craniofacial bone regeneration: A comprehensive review " submitted to [Biomolecules] Manuscript ID: biomolecules-2124234.

We have revised the manuscript following the recommendations of the editors and reviewers as outlined below:

Reviewer 1

Comment #1
“So for instance there are tissues for drug delivery https://doi.org/10.1021/acs.macromol.1c01596). What is the crucial difference between these and bone repair tissues?”

Answer

The authors really appreciate the reviewer’s comment. A new a paragraph was added under the section “2. Hydrogel scaffolds in bone regeneration” illustrating the difference between hydrogels used in drug delivery to tissues and those used in bone regeneration. Two new references were added including the recommended article.

Revised Text:

It is noteworthy, that various natural and synthetic polymeric based hydrogels have been designed for drug delivery. The combination of precise chemistry with multifunctional materials leads to unique responsive versatile hydrogels which can be employed as a potential platform to facilitate advanced biomedical applications. For example, amphiphilic linear pentablock hybrid polypeptides of the ABCBA type were synthesized using precise chemistry, where A is poly(L-lysine), B is poly(L-histidine)-co-poly(γ-benzyl-L-glutamate), and C is poly(ethylene oxide).The blocks' chain lengths were changed in order to produce hydrogels with various viscoelastic characteristics. An extrudable, in situ-forming, very quickly self-healing hydrogel with responsiveness to pH, temperature, and enzymes was produced. These characteristics would render the hydrogel appropriate for the directional and targeted delivery of cargo from the hydrogel toward cancer tissues in drug delivery since these tissues have lower pH and higher temperatures [31].  On the other hand, delivery systems that can safeguard various drugs, including osteoinductive growth factors from degradation, manage their delayed release to the intended site, and moderate their biological action for an extended time are required in the treatment and regeneration of damaged bone tissue [32].

Reviewer 2 Report

The review article is well-written with minor grammatical errors. However, a little more explanation about collagen/alginate-based hydrogel and nanogel might increase the article's impact on the readers.  

Author Response

Reviewer 2

Comment #1

 “The review article is well-written with minor grammatical errors. However, a little more explanation about collagen/alginate-based hydrogel and nanogel might increase the article's impact on the readers.”

Answer

According to the reviewer’s recommendation a new paragraph about collagen/alginate-based hydrogel was added in the “3.1.1.6. Collagen and alginate-based nanohydrogels” section.

Revised Text

Collagen and alginate can work together to combine their positive qualities and get over each material's drawbacks.  The weak mechanical properties of collagen and the inherent lack of cell-binding motifs inside alginate, which are the main hurdles that restrict their wide range of application can be overcome by their combination with upregulated cell-binding motifs and enhanced mechanical properties. Moreover, the ease of gelation of this composite under mild conditions enables the retention of bioactive agents and enhances cell encapsulation [154].  

Reviewer 3 Report

The manuscript entitled “Nano-based-polymeric hydrogels in craniofacial bone regeneration: A comprehensive review" by Karim M. Fawzy El-Sayed contains several interesting findings, and it may ultimately be suitable for publication. A significant effort was done to prepare the review article, which is quite interesting and unique. This article is impressive for the reviewer and audience of the biomedical community as well as the orthopedic research community. This article would show a significant impact on bone tissue engineering, nanomedicine, and the materials science community. There are some suggestions to improve the quality of the manuscript as it stands (detailed below) and these need to be addressed before it can be considered further. I thus recommend the paper be accepted after a minor revision.

The reviewer has the following comments  

1.       The title of the manuscript may be changed as follows

“Polymeric nanocomposite hydrogel scaffolds in craniofacial bone regeneration: A review”

2.       More Figures must be added to the revised manuscript

3.       Infographics need to be added to the revised manuscript

4.       The introduction section is very brief and may be improved entirely so that the reader can identify the scientific problems solved by this research. Moreover, the information on biomaterials may be elaborated on in the introduction with recent references (preferably 2021-2022). There are several biocompatible and biodegradable nanocomposites and they must be considered. Thus, the following article(s) may be quoted in the introduction and other relevant sections.

https://doi.org/10.1016/j.jmbbm.2021.104554, https://doi.org/10.3390/ph14111163, https://doi.org/10.1016/j.compositesb.2022.110150

In addition, The latest polymeric composites investigated by EV Barrera, Annabi, Seeram Ramakrishna, and Narsimha Mamidi, must be covered in the current review.

It would be more realistic to cover such kind of research work in the current manuscript. Which will enrich the quality of the current manuscript as well as the inquisitiveness of the readers.

Author Response

Reviewer 3

Comment #1

 “The title of the manuscript may be changed as follows “Polymeric nanocomposite hydrogel scaffolds in craniofacial bone regeneration: A review”.

Answer

According to the reviewer’s recommendation the title was changed.

Revised Title.

“Polymeric nanocomposite hydrogel scaffolds in craniofacial bone regeneration: A comprehensive review”

Comment #2
“More Figures must be added to the revised manuscript.”

Answer

The authors sincerely appreciate the reviewer’s comment, a figure in addition to three infographs illustrating the experiments involved in table 1,2 and 3 were added.

Comment#3

“Infographics need to be added to the revised manuscript”

Answer

The authors sincerely appreciate the reviewer’s comment, three infographs illustrating the experiments involved in table 1,2 and 3 were added.

Comment#4

“The introduction section is very brief and may be improved entirely so that the reader can identify the scientific problems solved by this research. Moreover, the information on biomaterials may be elaborated on in the introduction with recent references (preferably 2021-2022). There are several biocompatible and biodegradable nanocomposites and they must be considered. Thus, the following article(s) may be quoted in the introduction and other relevant sections.

https://doi.org/10.1016/j.jmbbm.2021.104554,

https://doi.org/10.3390/ph14111163,

https://doi.org/10.1016/j.compositesb.2022.110150.”

Answer

Authors greatly appreciates the reviewer’s comment. Two new paragraphs covering the recommended articles were added in (2.1.3.3. Hybrid hydrogels(. Three-dimensional printed hydrogels) section. In addition, a new paragraph was added in the ‘2.1.4.5. Three-dimensional printed hydrogels” section covering the recommended articles.

Revised Text

Carbonaceous nanomaterials (CNMs) have great potential among the nano-materials created with the development of nanotechnology and nanomedicine due to their significant usefulness. CNMs-based nanocomposites exceed other nanomaterials (metal, organic, etc.) in terms of surface immobilisation of macromolecules (such as: proteins, enzymes, peptides, etc.), bio-compatibility, mechanism of sensing, rapid transfer of electron kinetics ability, heat transfer, and surface adsorption ability thanks to their distinctive architectures, substantial surface area, ability to overcome biological barriers and impressive physicochemical characteristics. With the use of laser ablation, carbon vapour deposition, arc discharge, and joule heating, CNMs, which include fullerene, carbon nano-onions, carbon dots, graphene, graphene oxide, and reduced graphene oxide, are primarily manufactured [78].

Due to the strength qualities of carbon nanotube surfaces (CNTs), noncovalent interactions between polymer and CNT surfaces result in hybrid nanocomposite materials. Single wall carbon nanotubes (SWCNTs) in particular have been employed in biomedical field such as gene therapy, medical image processing, medication delivery, TE, and others [79]. A uni-compartmental knee implants with ultra-high molecular weight polyethylene sheets reinforced with functionalized SWCNTs with concentrations of 0.01 and 0.1 wt% was fabricated. The produced hybrid nanocomposite material samples exhibited improved yield strength, tensile strength, elongation, Young's modulus and after 14 days of incubation, human osteoblast cells demonstrated improved cell viability along with great cell growth and differentiation. Such combinations confirm the value of using CNTs for biomedical applications, providing an excellent potential for the creation of innovative composite biocompatible materials that can prolong human life [79].

To enhance medium percolation during 3D printing and, in turn, the proliferation of the cells attached to the material, the porosity of the material becomes crucial. Because of their superior biodegradability and biocompatibility, polyhydroxybutyrate (PHB)-based nanocomposites are frequently employed in both TE and drug delivery systems. However, due to insufficient physicochemical and mechanical qualities, PHB's utility in bone tissue engineering is restricted. Recently, PHB-based nanocomposites using a nanoblend and nano-clay with organically modified montmorillonite (MMT) were fabricated. The blended (PHB/MMT) has shown promise in 3D organs printing, lab-on-a-chip scaffold construction, and bone TE [95].